# The Convolution Theorem Involving Windowed Free Metaplectic Transform

**Manjun Cui** [1,2,3] and **Zhichao Zhang** [1,2,3,*]

1   School of Mathematics and Statistics, Nanjing University of Information Science and Technology, Nanjing 210044, China
2   Center for Applied Mathematics of Jiangsu Province, Nanjing University of Information Science and Technology, Nanjing 210044, China
3   Jiangsu International Joint Laboratory on System Modeling and Data Analysis, Nanjing University of Information Science and Technology, Nanjing 210044, China
*   Correspondence: zzc910731@163.com; Tel.: +86-13376073017

**Abstract:** The convolution product is widely used in many fields, such as signal processing, numerical analysis and so on; however, the convolution theorem in the domain of the windowed metaplectic transformation (WFMT) has not been studied. The primary goal of this paper is to give the convolution theorem of WFMT. Firstly, we review the definitions of the FMT and WFMT and give the inversion formula of the WFMT and the relationship between the FMT and WFMT. Then, according to the form of the classical convolution theorem and the convolution operator of the FMT, the convolution theorem in the domain of the WFMT is given. Finally, we prove the existence theorems of the proposed convolution theorem.

**Keywords:** free metaplectic transformation; windowed metaplectic transformation; convolution theorem; existence theorems





## 1. Introduction

The free metaplectic transformation (FMT), which is also known as a high-dimensional form of the nonseparable linear canonical transform (LCT), was first produced in [1]. A $2n \times 2n$ real symplectic matrix $\mathbf{M} = (\mathbf{A}, \mathbf{B}; \mathbf{C}, \mathbf{D})$ with $n(2n + 1)$ independent parameters was used in the free metaplectic transformation. When the symplectic matrix $\mathbf{M}$ takes different matrices, the FMT can reduce to different transformations, such as the Fourier transform (FT), the fractional Fourier transform (FRFT), the Fresnel transform (FRT), the linear canonical transform (LCT), and the basic quadratic phase factor multiplication [2]. Today, FMT is a central tool in time-frequency analysis, with several applications to PDE and mathematical physics [3–5]. Moreover, it has been widely used in optical systems, filter design, image processing, time-frequency analysis, harmonic analysis, and so on [6–9].

As a useful mathematical tool, the convolution product plays an important role in the design and implementation of multiplicative filters, harmonic analysis, image processing, and signal processing [10–12]. In recent years, people have conducted a lot of research on convolution theorems; many one-dimensional convolution theorems have been proposed, and many different transformations have been obtained based on convolution operators [13–16]. In the Fourier transform (FT) domain, the classical convolution theorem shows that the FT of two signals' convolution is equal to the product of their FT, which means that the FT can replace the complex convolution operation with a simple product operation. The FT has its own advantages and good effects in processing and analyzing stationary signals, but its ability in processing and analyzing nonstationary signals is weak due to its limitations. The fractional Fourier transform (FRFT) is an extension of the Fourier transform. By introducing a rotation angle $\alpha$, it can be considered that the signal rotates arbitrarily by angle $\alpha$ on the time axis, and the information of the time domain and frequency

domain can be obtained simultaneously [17,18]. In the fractional Fourier domain, a lot of work has been conducted on convolution; Wei proposed a new convolution structure for the FRFT [19]. On this basis, the paper [20] extended one dimension to high dimensions and gave the generalized fractional convolution theorem of $N$-dimension. The fractional Fourier transform uses a separable kernel and only has one extra degree of freedom, while the linear canonical transform (LCT) uses an inseparable kernel and introduces three extra degrees of freedom, which is more valuable for research. At present, various convolution operators and corresponding convolution theorems have been proposed in the LCT domain [21–25], which have made great contributions to signal processing and application. However, the FMT, as an $n$-dimensional nonseparable LCT, has higher extra degrees of freedom, which is the generalization of the LCT.

Although the FMT has been successfully applied in many fields, it has a large limitation. Because of its global kernel, there is no way to process the local spectral components of nontransient signals [26,27]. Therefore, in order to solve this defect, it is necessary to window the free metaplectic transformation; this concept was first put forward by Shah et al. in [28]. The windowed free metaplectic transformation (WFMT), also called the short time free metaplectic transformation, is an efficient signal processing tool that can locate the frequency spectrum of nontransient signals in the free metaplectic domain. The WFMT has more degrees of freedom and can produce a localized analysis of chirp signals that is effective.

In the windowed Fourier transform (WFT) domain, Lu and Zhang [29] used windowed convolution theorem for time-frequency analysis. In the windowed fractional Fourier (WFRFT) domain, Gao and Li developed a convolution form with a concise traditional convolution theorem in the frequency domain [30]. In the windowed linear canonical transform (WLCT) domain, the WLCT is defined by the convolution operator, and some properties and practical applications were studied in [31]. The WFT, WFRFT, and WFMT are all special forms of the WFMT, which urges us to actively explore the convolution theory of WFMT. Through the analysis of the present research situation, it is very meaningful to study the convolution theorem of the WFMT. Some scholars have studied the convolution theory of many time-frequency analysis tools, but no one has studied the convolution theory of WFMT; so, the study of the convolution theorem of WFMT is extremely important from a theoretical standpoint. To summarize, this paper studies the convolution theorem of the WFMT.

In this paper, we mainly study the convolution theorem of the WFMT. A convolution operator is proposed in the WFMT domain, which makes the convolution theorem in the WFMT domain have the same concise form as that in the traditional Fourier transform domain.In this way, we can transform the convolution in the time domain into the product in the metaplectic domain, which is very convenient and concise in the frequency domain and is beneficial to the design of the multiplication filter. Meanwhile, we prove the existence of the proposed convolution. The remainder of this paper is structured as follows. In Section 2, we briefly review the definition and elementary properties of the FMT. In Section 3, the definition of the WFMT and the related convolution theorem are given. In Section 4, we give the explanation of the existence theorem of the above convolution theorem. Finally, the conclusion is in Section 5.

## 2. Preliminaries

In this section, we mainly review some concepts and basic knowledge, which are used in the following sections.The space composed of all square integrable functions can be expressed as

$$L^2(\mathbb{R}^n) = \left\{ f : \int_{\mathbb{R}^n} |f(\mathbf{x})|^2 d\mathbf{x} < \infty \right\}. \tag{1}$$

For any positive integer $p$, the space formed by all $p$-integrable functions on $\mathbb{R}^n$ is expressed by $L^p(\mathbb{R}^n)$. For any function $f(\mathbf{x}) \in L^p(\mathbb{R}^n)$, the norm of $f$ is defined as

$$\| f \|_p = \left( \int_{\mathbb{R}^n} |f(\mathbf{x})|^p d\mathbf{x} \right)^{\frac{1}{p}}. \tag{2}$$

We denote a real symplectic $2n \times 2n$ matrix $\mathbf{M} = \begin{pmatrix} \mathbf{A} & \mathbf{B} \\ \mathbf{C} & \mathbf{D} \end{pmatrix}$ as $\mathbf{M} = (\mathbf{A}, \mathbf{B}; \mathbf{C}, \mathbf{D})$ (equivalently $det(\mathbf{B}) \neq 0$) for typographical convenience, where $\mathbf{A}, \mathbf{B}, \mathbf{C},$ and $\mathbf{D}$ are $n \times n$ real matrices satisfying

$$\mathbf{AB}^T = \mathbf{BA}^T, \mathbf{CD}^T = \mathbf{DC}^T, \mathbf{AD}^T - \mathbf{BC}^T = \mathbf{I_n}, \tag{3}$$

where $\mathbf{I_n}$ denotes the $n$-dimensional identity matrix. If $|det\mathbf{B}| \neq 0$ and $\mathbf{M}^T \mathbf{J} \mathbf{M} = \mathbf{J}$, the matrix $\mathbf{M} = (\mathbf{A}, \mathbf{B}; \mathbf{C}, \mathbf{D})$ is considered to be a free symplectic matrix. $\mathbf{M}^T = (\mathbf{A}^T, \mathbf{C}^T; \mathbf{B}^T, \mathbf{D}^T)$ and $\mathbf{M}^{-1} = (\mathbf{D}^T, -\mathbf{B}^T; -\mathbf{C}^T, \mathbf{A}^T)$ provide the transpose and the inverse of the free symplectic matrix $\mathbf{M} = (\mathbf{A}, \mathbf{B}; \mathbf{C}, \mathbf{D})$, respectively.

We define the FMT [32,33] as follows:

**Definition 1.** *The free metaplectic transform (FMT) of any function $f \in L^2(\mathbb{R}^n)$ with a free real symplectic matrix $\mathbf{M} = (\mathbf{A}, \mathbf{B}; \mathbf{C}, \mathbf{D})$ is denoted by $\mathcal{F}_\mathbf{M}[f]$ and is defined as [32,33]*

$$\mathcal{F}_\mathbf{M}[f](\mathbf{w}) = \int_{\mathbb{R}^n} f(\mathbf{x}) \mathcal{K}_\mathbf{M}(\mathbf{x}, \mathbf{w}) d\mathbf{x}, \tag{4}$$

*where kernel $\mathcal{K}_\mathbf{M}(\mathbf{x}, \mathbf{w})$ is given by*

$$\mathcal{K}_\mathbf{M}(\mathbf{x}, \mathbf{w}) = \frac{1}{\sqrt{det\mathbf{B}}} e^{i\pi(\mathbf{w}^T \mathbf{D} \mathbf{B}^{-1} \mathbf{w} - 2\mathbf{w}^T \mathbf{B}^{T-1} \mathbf{x} + \mathbf{x}^T \mathbf{B}^{-1} \mathbf{A} \mathbf{x})}, \tag{5}$$

*with $|det\mathbf{B}| \neq 0$.*

The corresponding (4) inversion formula is given by

$$f(\mathbf{x}) = \mathcal{F}_{\mathbf{M}^{-1}}[\mathcal{F}_\mathbf{M}[f](\mathbf{w})](\mathbf{x}) = \int_{\mathbb{R}^n} \mathcal{F}_\mathbf{M}[f](\mathbf{w}) \mathcal{K}_{\mathbf{M}^{-1}}(\mathbf{w}, \mathbf{x}) d\mathbf{w}. \tag{6}$$

## 3. Windowed Free Metaplectic Transform

**Definition 2** (WFMT)**.** *Let $g \in L^2(\mathbb{R}^n)$ be a window function; the windowed free metaplectic transform (WFMT) of any $f \in L^2(\mathbb{R}^n)$ with a free real symplectic matrix $\mathbf{M} = (\mathbf{A}, \mathbf{B}; \mathbf{C}, \mathbf{D})$ is defined as [28]*

$$(\mathcal{G}_g^\mathbf{M} f)(\mathbf{b}, \mathbf{u}) = \int_{\mathbb{R}^n} f(\mathbf{x}) \overline{g(\mathbf{x} - \mathbf{b})} \mathcal{K}_\mathbf{M}(\mathbf{x}, \mathbf{u}) d\mathbf{x}, \tag{7}$$

*where kernel $\mathcal{K}_\mathbf{M}(\mathbf{x}, \mathbf{u})$ is given as (5).*

**Remark 1.** *Let $\mathbf{A} = \mathbf{D} = diag(cos\alpha, cos\alpha, \ldots, cos\alpha)$ and $\mathbf{B} = -\mathbf{C} = diag(sin\alpha, sin\alpha, \ldots, sin\alpha)$; then, the definition of the WFMT will reduce to the windowed fractional Fourier transform (WFRFT) with rotational angle $\alpha$.*

**Remark 2.** *When the window function $g(\mathbf{x}) = 1$, the definition of the WFMT [28] equals the FMT [32,33].*

From (4) and (7), the relationship between the FMT and WFMT can be written as

$$(\mathcal{G}_g^\mathbf{M} f)(\mathbf{b}, \mathbf{u}) = \mathcal{F}_\mathbf{M}[h](\mathbf{u}), \tag{8}$$

where $h(\mathbf{x}, \mathbf{b}) = f(\mathbf{x}) \overline{g(\mathbf{x} - \mathbf{b})}$.

**Proposition 1** (The inversion of the WFMT). *Let $g \in L^2(\mathbb{R}^n)$ be a window function, and $\|g\|_2 \neq 0$; $\mathcal{G}_g^{\mathbf{M}} f$ denotes the windowed free metaplectic transform of $f$, and*

$$f(\mathbf{x}) = \frac{1}{\|g\|_2} \int_{\mathbb{R}^n} \int_{\mathbb{R}^n} (\mathcal{G}_g^{\mathbf{M}} f)(\mathbf{b}, \mathbf{u}) \overline{\mathcal{K}_{\mathbf{M}}(\mathbf{x}, \mathbf{u})} g(\mathbf{x} - \mathbf{b}) d\mathbf{u} d\mathbf{b}. \tag{9}$$

**Proof of Proposition 1.** Letting $h(\mathbf{x}, \mathbf{b}) = f(\mathbf{x}) \overline{g(\mathbf{x} - \mathbf{b})}$,

$$(\mathcal{G}_g^{\mathbf{M}} f)(\mathbf{b}, \mathbf{u}) = \int_{\mathbb{R}^n} h(\mathbf{x}, \mathbf{b}) \mathcal{K}_{\mathbf{M}}(\mathbf{x}, \mathbf{u}) d\mathbf{x} = \mathcal{F}_{\mathbf{M}}[h](\mathbf{u}). \tag{10}$$

According to the inversion formula of FMT (6), we have

$$h(\mathbf{x}, \mathbf{b}) = \int_{\mathbb{R}^n} \mathcal{F}_{\mathbf{M}}[h](\mathbf{u}) \mathcal{K}_{\mathbf{M}^{-1}}(\mathbf{u}, \mathbf{x}) d\mathbf{u}. \tag{11}$$

Then,

$$f(\mathbf{x}) \overline{g(\mathbf{x} - \mathbf{b})} = \int_{\mathbb{R}^n} (\mathcal{G}_g^{\mathbf{M}} f)(\mathbf{b}, \mathbf{u}) \overline{\mathcal{K}_{\mathbf{M}}(\mathbf{x}, \mathbf{u})} d\mathbf{u}. \tag{12}$$

Multiplying $g(\mathbf{x} - \mathbf{b})$ on both sides of the above equation and integrating, we obtain

$$\int_{\mathbb{R}^n} f(\mathbf{x}) \overline{g(\mathbf{x} - \mathbf{b})} g(\mathbf{x} - \mathbf{b}) d\mathbf{b} = \int_{\mathbb{R}^n} \int_{\mathbb{R}^n} (\mathcal{G}_g^{\mathbf{M}} f)(\mathbf{b}, \mathbf{u}) \overline{\mathcal{K}_{\mathbf{M}}(\mathbf{x}, \mathbf{u})} g(\mathbf{x} - \mathbf{b}) d\mathbf{u} d\mathbf{b}. \tag{13}$$

That is,

$$f(x) \|g\|_2^2 = \int_{\mathbb{R}^n} \int_{\mathbb{R}^n} (\mathcal{G}_g^{\mathbf{M}} f)(\mathbf{b}, \mathbf{u}) \overline{\mathcal{K}_{\mathbf{M}}(\mathbf{x}, \mathbf{u})} g(\mathbf{x} - \mathbf{b}) d\mathbf{u} d\mathbf{b}. \tag{14}$$

Since $\|g\|_2 \neq 0$, we can obtain

$$f(x) = \frac{1}{\|g\|_2} \int_{\mathbb{R}^n} \int_{\mathbb{R}^n} (\mathcal{G}_g^{\mathbf{M}} f)(\mathbf{b}, \mathbf{u}) \overline{\mathcal{K}_{\mathbf{M}}(\mathbf{x}, \mathbf{u})} g(\mathbf{x} - \mathbf{b}) d\mathbf{u} d\mathbf{b}.$$

Thus, the proof is finished. □

**Definition 3.** *Given two functions $f, h \in L^2(\mathbb{R}^n)$, the free metaplectic convolution with a free symplectic matrix $\mathbf{M} = (\mathbf{A}, \mathbf{B}; \mathbf{C}, \mathbf{D})$ is denoted by $\circledast_{\mathbf{M}}$ and is defined as [32]*

$$(f \circledast_{\mathbf{M}} h)(\mathbf{b}) = \frac{1}{\sqrt{det\mathbf{B}}} \int_{\mathbb{R}^n} f(\mathbf{x}) h(\mathbf{b} - \mathbf{x}) e^{i\pi \left( \mathbf{x}^T \mathbf{B}^{-1} \mathbf{A}(\mathbf{x} - \mathbf{b}) + (\mathbf{x} - \mathbf{b})^T \mathbf{B}^{-1} \mathbf{A} \mathbf{x} \right)} d\mathbf{x}. \tag{15}$$

According to (15), $f \circledast_M h$ can be written as

$$(f \circledast_{\mathbf{M}} h)(\mathbf{b}) = \frac{1}{\sqrt{det\mathbf{B}}} (f_1 * h_1)(\mathbf{b}), \tag{16}$$

where $f_1(\mathbf{x}) = f(\mathbf{x}) e^{i\pi \left( \mathbf{x}^T \mathbf{B}^{-1} \mathbf{A}(\mathbf{x} - \mathbf{b}) \right)}$, $h_1(\mathbf{x}) = h(\mathbf{x}) e^{i\pi \left( \mathbf{x}^T \mathbf{B}^{-1} \mathbf{A}(\mathbf{x} - \mathbf{b}) \right)}$, and $*$ denotes the common convolution operator.

**Proposition 2.** *For two functions $f, h \in L^2(\mathbb{R}^n)$ in free metaplectic domains, the convolution theorem can be written as*

$$\mathcal{F}_{\mathbf{M}}[(f \circledast_{\mathbf{M}} h)(\mathbf{b})](\mathbf{w}) = e^{-i\pi \mathbf{w}^T \mathbf{D} \mathbf{B}^{-1} \mathbf{w}} \mathcal{F}_{\mathbf{M}}[f](\mathbf{w}) \mathcal{F}_{\mathbf{M}}[h](\mathbf{w}). \tag{17}$$

**Proof of Proposition 2.** The proof is very simple; so, it is omitted here. □

According to (7) and (15), the WFMT can be expressed in terms of the convolution

$$(\mathcal{G}_g^{\mathbf{M}} f)(\mathbf{b}, \mathbf{u}) = \left( \widetilde{f}_{\mathbf{u}} \circledast_{\mathbf{M}} \widetilde{g_{\mathbf{b}}} \right)(\mathbf{b}), \tag{18}$$

where

$$\widetilde{f}_{\mathbf{u}}(\mathbf{x}) = f(\mathbf{x}) e^{i\pi(\mathbf{u}^T \mathbf{D}\mathbf{B}^{-1}\mathbf{u} - 2\mathbf{u}^T \mathbf{B}^{T^{-1}}\mathbf{x} + \mathbf{x}^T \mathbf{B}^{-1}\mathbf{A}\mathbf{x})}, \tag{19}$$

$$\widetilde{g_{\mathbf{b}}}(\mathbf{x}) = \overline{g(-\mathbf{x})} e^{i\pi\left( (\mathbf{b}-\mathbf{x})^T \mathbf{B}^{-1}\mathbf{A}\mathbf{x} + \mathbf{x}^T \mathbf{B}^{-1}\mathbf{A}(\mathbf{b}-\mathbf{x}) \right)}. \tag{20}$$

We define the convolution product associated with the WFMT as

$$\mathcal{G}_g^{\mathbf{M}}(f \star_{\mathbf{M}} h)(\mathbf{b}, \mathbf{u}) = \left( \mathcal{G}_g^{\mathbf{M}} f \right)(\mathbf{b}, \mathbf{u}) \left( \mathcal{G}_g^{\mathbf{M}} h \right)(\mathbf{b}, \mathbf{u}). \tag{21}$$

**Theorem 1** (Convolution theorem for the WFMT). *Let $g \in L^2(\mathbb{R}^n)$ be a window function, $f \in L^2(\mathbb{R}^n)$ and $h \in L^2(\mathbb{R}^n)$. For any $\mathbf{x} \in \mathbb{R}^n$, the windowed convolution is obtained by*

$$(f \star_{\mathbf{M}} h)(\mathbf{x}) = \int_{\mathbb{R}^n} \int_{\mathbb{R}^n} A_{\mathbf{u}}(\boldsymbol{\xi}, \mathbf{y}, \mathbf{x}) f(\boldsymbol{\xi}) h(\mathbf{y}) d\boldsymbol{\xi} \, d\mathbf{y}, \tag{22}$$

*where*

$$
\begin{aligned}
A_{\mathbf{u}}(\boldsymbol{\xi}, \mathbf{y}, \mathbf{x}) = &\frac{1}{|det\mathbf{B}|} \int_{\mathbb{R}^n} \int_{\mathbb{R}^n} \overline{\mathcal{K}_{\mathbf{M}}(\mathbf{x}, \mathbf{w})} \mathcal{K}_{\mathbf{M}}(\mathbf{t}, \mathbf{w}) \frac{U(\boldsymbol{\xi}, \mathbf{u})U(\mathbf{y}, \mathbf{u})}{U(\mathbf{x}, \mathbf{u})} \\
&\times \frac{\widetilde{g_{\mathbf{b}}}(\mathbf{t} - \boldsymbol{\xi}) \widetilde{g_{\mathbf{b}}}(\mathbf{t} - \mathbf{y})}{\mathcal{F}_{\mathbf{M}}[\widetilde{g_{\mathbf{b}}}](\mathbf{w})} e^{i\pi\left( \boldsymbol{\xi}^T \mathbf{B}^{-1}\mathbf{A}(\boldsymbol{\xi}-\mathbf{t}) + (\boldsymbol{\xi}-\mathbf{t})^T \mathbf{B}^{-1}\mathbf{A}\boldsymbol{\xi} \right)} \\
&\times e^{i\pi\left( \mathbf{y}^T \mathbf{B}^{-1}\mathbf{A}(\mathbf{y}-\mathbf{t}) + (\mathbf{y}-\mathbf{t})^T \mathbf{B}^{-1}\mathbf{A}\mathbf{y} \right)} e^{i\pi \mathbf{w}^T \mathbf{D}\mathbf{B}^{-1}\mathbf{w}} d\mathbf{t} d\mathbf{w},
\end{aligned} \tag{23}
$$

$$U(\mathbf{x}, \mathbf{u}) = e^{i\pi(\mathbf{u}^T \mathbf{D}\mathbf{B}^{-1}\mathbf{u} - 2\mathbf{u}^T \mathbf{B}^{T^{-1}}\mathbf{x} + \mathbf{x}^T \mathbf{B}^{-1}\mathbf{A}\mathbf{x})}, \tag{24}$$

$$\widetilde{g_{\mathbf{b}}}(\mathbf{x}) = \overline{g(-\mathbf{x})} e^{i\pi\left( (\mathbf{b}-\mathbf{x})^T \mathbf{B}^{-1}\mathbf{A}\mathbf{x} + \mathbf{x}^T \mathbf{B}^{-1}\mathbf{A}(\mathbf{b}-\mathbf{x}) \right)}. \tag{25}$$

**Proof of Theorem 1.** According to (17) and (18), we have

$$
\begin{aligned}
\mathcal{F}_{\mathbf{M}}\left[ (\mathcal{G}_g^{\mathbf{M}} f)(\mathbf{b}, \mathbf{u}) \right](\mathbf{w}) &= \mathcal{F}_{\mathbf{M}}\left[ \left( \widetilde{f}_{\mathbf{u}} \circledast_{\mathbf{M}} \widetilde{g_{\mathbf{b}}} \right)(\mathbf{b}) \right](\mathbf{w}) \\
&= e^{-i\pi \mathbf{w}^T \mathbf{D}\mathbf{B}^{-1}\mathbf{w}} \mathcal{F}_{\mathbf{M}}[\widetilde{f}_{\mathbf{u}}](\mathbf{w}) \mathcal{F}_{\mathbf{M}}[\widetilde{g_{\mathbf{b}}}](\mathbf{w}).
\end{aligned} \tag{26}
$$

Let $U(\mathbf{x}, \mathbf{u}) = e^{i\pi(\mathbf{u}^T \mathbf{D}\mathbf{B}^{-1}\mathbf{u} - 2\mathbf{u}^T \mathbf{B}^{T^{-1}}\mathbf{x} + \mathbf{x}^T \mathbf{B}^{-1}\mathbf{A}\mathbf{x})}$; then, we have

$$
\begin{aligned}
&\mathcal{F}_{\mathbf{M}}\left[ (\mathcal{G}_g^{\mathbf{M}}(f \star_{\mathbf{M}} h))(\mathbf{b}, \mathbf{u}) \right](\mathbf{w}) \\
&= e^{-i\pi \mathbf{w}^T \mathbf{D}\mathbf{B}^{-1}\mathbf{w}} \mathcal{F}_{\mathbf{M}}[\widetilde{(f \star_{\mathbf{M}} h)}_{\mathbf{u}}](\mathbf{w}) \mathcal{F}_{\mathbf{M}}[\widetilde{g_{\mathbf{b}}}](\mathbf{w}) \\
&= e^{-i\pi \mathbf{w}^T \mathbf{D}\mathbf{B}^{-1}\mathbf{w}} \mathcal{F}_{\mathbf{M}}[(f \star_{\mathbf{M}} h)(\mathbf{x})U(\mathbf{x}, \mathbf{u})](\mathbf{w}) \mathcal{F}_{\mathbf{M}}[\widetilde{g_{\mathbf{b}}}](\mathbf{w}).
\end{aligned} \tag{27}
$$

Moreover, we have

$$
\begin{aligned}
\mathcal{F}_{\mathbf{M}}\left[ (\mathcal{G}_g^{\mathbf{M}}(f \star_{\mathbf{M}} h))(\mathbf{b}, \mathbf{u}) \right](\mathbf{w}) &= \mathcal{F}_{\mathbf{M}}\left[ \left( \mathcal{G}_g^{\mathbf{M}} f \right)(\mathbf{b}, \mathbf{u}) \left( \mathcal{G}_g^{\mathbf{M}} h \right)(\mathbf{b}, \mathbf{u}) \right](\mathbf{w}) \\
&= \mathcal{F}_{\mathbf{M}}\left[ \left( \widetilde{f}_{\mathbf{u}} \circledast_{\mathbf{M}} \widetilde{g_{\mathbf{b}}} \right) \left( \widetilde{h}_{\mathbf{u}} \circledast_{\mathbf{M}} \widetilde{g_{\mathbf{b}}} \right) \right](\mathbf{w}).
\end{aligned} \tag{28}
$$

Thus,

$$
\begin{aligned}
&e^{-i\pi \mathbf{w}^T \mathbf{DB}^{-1}\mathbf{w}} \mathcal{F}_{\mathbf{M}}[(f \star_{\mathbf{M}} h)(\mathbf{x})U(\mathbf{x},\mathbf{u})](\mathbf{w}) \mathcal{F}_{\mathbf{M}}[\widetilde{g_{\mathbf{b}}}](\mathbf{w}) \\
&= \mathcal{F}_{\mathbf{M}}\left[ \left( \widetilde{f_{\mathbf{u}}} \circledast_{\mathbf{M}} \widetilde{g_{\mathbf{b}}} \right) \left( \widetilde{h_{\mathbf{u}}} \circledast_{\mathbf{M}} \widetilde{g_{\mathbf{b}}} \right) \right](\mathbf{w}).
\end{aligned}
\tag{29}
$$

So,

$$
\begin{aligned}
&\mathcal{F}_{\mathbf{M}}[(f \star_{\mathbf{M}} h)(\mathbf{x})U(\mathbf{x},\mathbf{u})](\mathbf{w}) \\
&= e^{i\pi \mathbf{w}^T \mathbf{DB}^{-1}\mathbf{w}} \frac{\mathcal{F}_{\mathbf{M}}\left[ \left( \widetilde{f_{\mathbf{u}}} \circledast_{\mathbf{M}} \widetilde{g_{\mathbf{b}}} \right) \left( \widetilde{h_{\mathbf{u}}} \circledast_{\mathbf{M}} \widetilde{g_{\mathbf{b}}} \right) \right](\mathbf{w})}{\mathcal{F}_{\mathbf{M}}[\widetilde{g_{\mathbf{b}}}](\mathbf{w})}.
\end{aligned}
\tag{30}
$$

According to the inversion Formula (6), we can obtain

$$
\begin{aligned}
(f \star_{\mathbf{M}} h)(\mathbf{x})U(\mathbf{x},\mathbf{u}) =& \mathcal{F}_{\mathbf{M}}^{-1}\left[ e^{i\pi \mathbf{w}^T \mathbf{DB}^{-1}\mathbf{w}} \frac{\mathcal{F}_{\mathbf{M}}\left[ \left( \widetilde{f_{\mathbf{u}}} \circledast_{\mathbf{M}} \widetilde{g_{\mathbf{b}}} \right) \left( \widetilde{h_{\mathbf{u}}} \circledast_{\mathbf{M}} \widetilde{g_{\mathbf{b}}} \right) \right](\mathbf{w})}{\mathcal{F}_{\mathbf{M}}[\widetilde{g_{\mathbf{b}}}](\mathbf{w})} \right] \\
=& \int_{\mathbb{R}^n} \frac{e^{i\pi \mathbf{w}^T \mathbf{DB}^{-1}\mathbf{w}}}{\mathcal{F}_{\mathbf{M}}[\widetilde{g_{\mathbf{b}}}](\mathbf{w})} \mathcal{F}_{\mathbf{M}}\left[ \left( \widetilde{f_{\mathbf{u}}} \circledast_{\mathbf{M}} \widetilde{g_{\mathbf{b}}} \right) \left( \widetilde{h_{\mathbf{u}}} \circledast_{\mathbf{M}} \widetilde{g_{\mathbf{b}}} \right) \right](\mathbf{w}) \overline{\mathcal{K}_{\mathbf{M}}(\mathbf{x},\mathbf{w})} d\mathbf{w} \\
=& \int_{\mathbb{R}^n} \frac{e^{i\pi \mathbf{w}^T \mathbf{DB}^{-1}\mathbf{w}}}{\mathcal{F}_{\mathbf{M}}[\widetilde{g_{\mathbf{b}}}](\mathbf{w})} \left( \int_{\mathbb{R}^n} \left( \widetilde{f_{\mathbf{u}}} \circledast_{\mathbf{M}} \widetilde{g_{\mathbf{b}}} \right)(\mathbf{t}) \left( \widetilde{h_{\mathbf{u}}} \circledast_{\mathbf{M}} \widetilde{g_{\mathbf{b}}} \right)(\mathbf{t}) \mathcal{K}_{\mathbf{M}}(\mathbf{t},\mathbf{w}) d\mathbf{t} \right) \\
& \times \overline{\mathcal{K}_{\mathbf{M}}(\mathbf{x},\mathbf{w})} d\mathbf{w} \\
=& \frac{1}{|det\mathbf{B}|} \int_{\mathbb{R}^n} \int_{\mathbb{R}^n} \int_{\mathbb{R}^n} \int_{\mathbb{R}^n} \frac{e^{i\pi \mathbf{w}^T \mathbf{DB}^{-1}\mathbf{w}}}{\mathcal{F}_{\mathbf{M}}[\widetilde{g_{\mathbf{b}}}](\mathbf{w})} \mathcal{K}_{\mathbf{M}}(\mathbf{t},\mathbf{w}) \overline{\mathcal{K}_{\mathbf{M}}(\mathbf{x},\mathbf{w})} \\
& \times \widetilde{f_{\mathbf{u}}}(\boldsymbol{\xi}) \widetilde{g_{\mathbf{b}}}(\mathbf{t}-\boldsymbol{\xi}) e^{i\pi \left( \boldsymbol{\xi}^T \mathbf{B}^{-1}\mathbf{A}(\boldsymbol{\xi}-\mathbf{t}) + (\boldsymbol{\xi}-\mathbf{t})^T B^{-1} \mathbf{A}\boldsymbol{\xi} \right)} \\
& \times \widetilde{h_{\mathbf{u}}}(\mathbf{y}) \widetilde{g_{\mathbf{b}}}(\mathbf{t}-\mathbf{y}) e^{i\pi \left( \mathbf{y}^T \mathbf{B}^{-1}\mathbf{A}(\mathbf{y}-\mathbf{t}) + (\mathbf{y}-\mathbf{t})^T \mathbf{B}^{-1}\mathbf{A}\mathbf{y} \right)} d\boldsymbol{\xi} d\mathbf{y} d\mathbf{w} d\mathbf{t} \\
=& \frac{1}{|det\mathbf{B}|} \int_{\mathbb{R}^n} \int_{\mathbb{R}^n} \int_{\mathbb{R}^n} \int_{\mathbb{R}^n} \mathcal{K}_{\mathbf{M}}(\mathbf{t},\mathbf{w}) \overline{\mathcal{K}_{\mathbf{M}}(\mathbf{x},\mathbf{w})} \\
& \times \frac{\widetilde{g_{\mathbf{b}}}(\mathbf{t}-\boldsymbol{\xi}) \widetilde{g_{\mathbf{b}}}(\mathbf{t}-\mathbf{y})}{\mathcal{F}_{\mathbf{M}}[\widetilde{g_{\mathbf{b}}}](\mathbf{w})} e^{i\pi \left( \boldsymbol{\xi}^T \mathbf{B}^{-1}\mathbf{A}(\boldsymbol{\xi}-\mathbf{t}) + (\boldsymbol{\xi}-\mathbf{t})^T \mathbf{B}^{-1}\mathbf{A}\boldsymbol{\xi} \right)} \\
& \times e^{i\pi \left( \mathbf{y}^T \mathbf{B}^{-1}\mathbf{A}(\mathbf{y}-\mathbf{t}) + (\mathbf{y}-\mathbf{t})^T \mathbf{B}^{-1}\mathbf{A}\mathbf{y} \right)} e^{i\pi \mathbf{w}^T \mathbf{DB}^{-1}\mathbf{w}} f(\boldsymbol{\xi})U(\boldsymbol{\xi},\mathbf{u}) \\
& \times h(\mathbf{y})U(\mathbf{y},\mathbf{u}) d\boldsymbol{\xi} d\mathbf{y} d\mathbf{w} d\mathbf{t}.
\end{aligned}
\tag{31}
$$

Since $U(\mathbf{x},\mathbf{u}) \neq 0$, we can obtain

$$
(f \star_{\mathbf{M}} h)(\mathbf{x}) = \int_{\mathbb{R}^n} \int_{\mathbb{R}^n} A_{\mathbf{u}}(\boldsymbol{\xi},\mathbf{y},\mathbf{x}) f(\boldsymbol{\xi}) h(\mathbf{y}) d\boldsymbol{\xi} \, d\mathbf{y},
$$

where $A_{\mathbf{u}}(\boldsymbol{\xi},\mathbf{y},\mathbf{x})$ is defined as (23). Thus, the proof is finished. □

**Remark 3.** *Let* $\mathbf{A} = \mathbf{D} = diag(cos\alpha, cos\alpha, \dots, cos\alpha)$ *and* $\mathbf{B} = -\mathbf{C} = diag(sin\alpha, sin\alpha, \dots, sin\alpha)$ *in the matrix* $\mathbf{M} = (\mathbf{A}, \mathbf{B}; \mathbf{C}, \mathbf{D})$; *then, the convolution theorem of the WFMT will reduce to the convolution theorem of the WFRFT in [30].*

**Remark 4.** *Since the FMT uses the global kernel, there is no way to deal with the local spectral components of nontransient signals, but the WFMT can solve this problem well. Therefore, the convolution theorem proposed in this paper is less limited than the convolution theorem proposed in [32], and it is better for capturing and analyzing nontransient signals. The convolution theorem (21) has the simplicity and elegance consistent with the classical convolution theorem in the FT. However, the convolution theorems in [28,32,33] are not so concise and elegant with an*

*extra chirp function. Through the convolution theorem proposed in this paper, we can transform the convolution in the time domain into the simple product in the metaplectic domain.*

**Theorem 2.** *For any functions $f, h, k \in L^2(\mathbb{R}^n)$ and the free real symplectic matrix $\mathbf{M}$, the convolution of the WFMT satisfies the commutative law and the associative law:*
*(i) (Commutative law)*

$$f \star_{\mathbf{M}} h = h \star_{\mathbf{M}} f. \tag{32}$$

*(ii) (Associative law)*

$$(f \star_{\mathbf{M}} h) \star_{\mathbf{M}} k = f \star_{\mathbf{M}} (h \star_{\mathbf{M}} k). \tag{33}$$

**Proof of Theorem 2.** The commutative law is obvious. Next, we prove the associative law.

$$
\begin{aligned}
((f \star_{\mathbf{M}} h) \star_{\mathbf{M}} k)(\mathbf{m}) &= \int_{\mathbb{R}^n} \int_{\mathbb{R}^n} A_{\mathbf{u}}(\mathbf{x}, \mathbf{l}, \mathbf{m})(f \star_{\mathbf{M}} h)(\mathbf{x}) k(\mathbf{l}) d\mathbf{x} d\mathbf{l} \\
&= \int_{\mathbb{R}^n} \int_{\mathbb{R}^n} A_{\mathbf{u}}(\mathbf{x}, \mathbf{l}, \mathbf{m}) k(\mathbf{l}) \int_{\mathbb{R}^n} \int_{\mathbb{R}^n} A_{\mathbf{u}}(\boldsymbol{\xi}, \mathbf{y}, \mathbf{x}) f(\boldsymbol{\xi}) h(\mathbf{y}) d\boldsymbol{\xi} d\mathbf{y} d\mathbf{x} d\mathbf{l} \\
&= \int_{\mathbb{R}^n} \int_{\mathbb{R}^n} \int_{\mathbb{R}^n} \int_{\mathbb{R}^n} A_{\mathbf{u}}(\mathbf{x}, \mathbf{l}, \mathbf{m}) A_{\mathbf{u}}(\boldsymbol{\xi}, \mathbf{y}, \mathbf{x}) f(\boldsymbol{\xi}) g(\mathbf{y}) k(\mathbf{l}) d\mathbf{x} d\mathbf{y} d\mathbf{l} d\boldsymbol{\xi},
\end{aligned} \tag{34}
$$

$$
\begin{aligned}
(f \star_{\mathbf{M}} (h \star_{\mathbf{M}} k))(\mathbf{m}) &= \int_{\mathbb{R}^n} \int_{\mathbb{R}^n} A_{\mathbf{u}}(\boldsymbol{\xi}, \mathbf{x}, \mathbf{m})(h \star_{\mathbf{M}} k)(\mathbf{x}) f(\boldsymbol{\xi}) d\boldsymbol{\xi} d\mathbf{x} \\
&= \int_{\mathbb{R}^n} \int_{\mathbb{R}^n} A_{\mathbf{u}}(\boldsymbol{\xi}, \mathbf{x}, \mathbf{m}) f(\boldsymbol{\xi}) \int_{\mathbb{R}^n} \int_{\mathbb{R}^n} A_{\mathbf{u}}(\mathbf{y}, \mathbf{l}, \mathbf{x}) h(\mathbf{y}) k(\mathbf{l}) d\mathbf{y} d\mathbf{l} d\boldsymbol{\xi} d\mathbf{x} \\
&= \int_{\mathbb{R}^n} \int_{\mathbb{R}^n} \int_{\mathbb{R}^n} \int_{\mathbb{R}^n} A_{\mathbf{u}}(\boldsymbol{\xi}, \mathbf{x}, \mathbf{m}) A_{\mathbf{u}}(\mathbf{y}, \mathbf{l}, \mathbf{x}) f(\boldsymbol{\xi}) h(\mathbf{y}) k(\mathbf{l}) d\mathbf{x} d\mathbf{y} d\mathbf{l} d\boldsymbol{\xi}.
\end{aligned} \tag{35}
$$

Obviously, $A_{\mathbf{u}}(\mathbf{x}, \mathbf{l}, \mathbf{m}) A_{\mathbf{u}}(\boldsymbol{\xi}, \mathbf{y}, \mathbf{x}) = A_{\mathbf{u}}(\boldsymbol{\xi}, \mathbf{x}, \mathbf{m}) A_{\mathbf{u}}(\mathbf{y}, \mathbf{l}, \mathbf{x})$. Thus,

$$(f \star_{\mathbf{M}} h) \star_{\mathbf{M}} k = f \star_{\mathbf{M}} (h \star_{\mathbf{M}} k).$$

Thus, the proof is finished. □

**Example 1.** *For simplicity, we chose a two-dimensional Gaussian function $g(\mathbf{x}) = e^{-(x_1{}^2 + x_2{}^2)}$ as the window function and fixed the center of the window function at the origin. For the symplectic matrix $\mathbf{M}$, we chose*

$$
\mathbf{M} = \begin{pmatrix} \mathbf{A} & \mathbf{B} \\ \mathbf{C} & \mathbf{D} \end{pmatrix} = \begin{pmatrix} 1 & 2 & -1/2 & -1/2 \\ -2 & 1 & -1/2 & 1/2 \\ -1 & -3 & 1 & 1 \\ -1 & 0 & -1/2 & 1/2 \end{pmatrix}. \tag{36}
$$

*Thus,*

$$
\mathbf{B}^{-1} = \begin{pmatrix} -1 & -1 \\ -1 & 1 \end{pmatrix},
$$

$$
\mathbf{w}^T \mathbf{D} \mathbf{B}^{-1} \mathbf{w} = \begin{pmatrix} w_1 & w_2 \end{pmatrix} \begin{pmatrix} -2 & 0 \\ 0 & 1 \end{pmatrix} \begin{pmatrix} w_1 \\ w_2 \end{pmatrix} = -2w_1{}^2 + w_2{}^2, \tag{37}
$$

$$\mathbf{w}^T \mathbf{B}^{T^{-1}} \mathbf{x} = \begin{pmatrix} w_1 & w_2 \end{pmatrix} \begin{pmatrix} -1 & -1 \\ -1 & 1 \end{pmatrix} \begin{pmatrix} x_1 \\ x_2 \end{pmatrix} = -w_1 x_2 - w_1 x_2 - w_2 x_1 + w_2 x_2, \quad (38)$$

$$\mathbf{x}^T \mathbf{B}^{-1} \mathbf{A} \mathbf{x} = \begin{pmatrix} x_1 & x_2 \end{pmatrix} \begin{pmatrix} 1 & -3 \\ -3 & -1 \end{pmatrix} \begin{pmatrix} x_1 \\ x_2 \end{pmatrix} = x_1{}^2 - 6 x_1 x_2 - x_2{}^2, \quad (39)$$

$$\mathbf{x}^T \mathbf{B}^{-1} \mathbf{A} \mathbf{y} = \begin{pmatrix} x_1 & x_2 \end{pmatrix} \begin{pmatrix} 1 & -3 \\ -3 & -1 \end{pmatrix} \begin{pmatrix} y_1 \\ y_2 \end{pmatrix} = (x_1 - 3 x_2) y_1 + (3 x_1 + x_2) y_2, \quad (40)$$

$$\mathcal{K}_{\mathbf{M}}(\mathbf{t}, \mathbf{w}) = \sqrt{2} e^{i\pi\left(\left(-2w_1{}^2 + w_2{}^2\right) + 2(w_1 t_2 + w_1 t_2 + w_2 t_1 - w_2 t_2) + \left(t_1{}^2 - 6 t_1 t_2 - t_2{}^2\right)\right)}, \quad (41)$$

$$U(\mathbf{x}, \mathbf{u}) = e^{i\pi\left(\left(-2u_1{}^2 + u_2{}^2\right) + 2(u_1 x_2 + u_1 x_2 + u_2 x_1 - u_2 x_2) + \left(x_1{}^2 - 6 x_1 x_2 - x_2{}^2\right)\right)}, \quad (42)$$

$$\widetilde{g_{\mathbf{b}}}(\mathbf{x}) = e^{-\left(x_1{}^2 + x_2{}^2\right)} e^{-2i\pi\left(x_1{}^2 - 6 x_1 x_2 - x_2{}^2\right)}. \quad (43)$$

*According to the above formulas, we obtain*

$$
\begin{aligned}
(f \star_{\mathbf{M}} h)(\mathbf{x}) &= \int_{\mathbb{R}^2} \int_{\mathbb{R}^2} A_{\mathbf{u}}(\boldsymbol{\xi}, \mathbf{y}, \mathbf{x}) f(\boldsymbol{\xi}) h(\mathbf{y}) d\boldsymbol{\xi} \, d\mathbf{y} \\
&= 2 \int_{\mathbb{R}^2} \int_{\mathbb{R}^2} \int_{\mathbb{R}^2} \int_{\mathbb{R}^2} \sqrt{2} e^{i\pi\left(\left(-2w_1{}^2 + w_2{}^2\right) + 2(w_1 t_2 + w_1 t_2 + w_2 t_1 - w_2 t_2) + \left(t_1{}^2 - 6 t_1 t_2 - t_2{}^2\right)\right)} \\
&\quad \times \sqrt{2} e^{-i\pi\left(\left(-2w_1{}^2 + w_2{}^2\right) + 2(w_1 x_2 + w_1 x_2 + w_2 x_1 - w_2 x_2) + \left(x_1{}^2 - 6 x_1 x_2 - x_2{}^2\right)\right)} \\
&\quad \times e^{-i\pi\left(\left(-2u_1{}^2 + u_2{}^2\right) + 2(u_1 x_2 + u_1 x_2 + u_2 x_1 - u_2 x_2) + \left(x_1{}^2 - 6 x_1 x_2 - x_2{}^2\right)\right)} \\
&\quad \times e^{i\pi\left(\left(-2u_1{}^2 + u_2{}^2\right) + 2(u_1 \xi_2 + u_1 \xi_2 + u_2 \xi_1 - u_2 \xi_2) + \left(\xi_1{}^2 - 6 \xi_1 \xi_2 - \xi_2{}^2\right)\right)} \\
&\quad \times e^{i\pi\left(\left(-2u_1{}^2 + u_2{}^2\right) + 2(u_1 y_2 + u_1 y_2 + u_2 y_1 - u_2 y_2) + \left(y_1{}^2 - 6 y_1 y_2 - y_2{}^2\right)\right)} \\
&\quad \times e^{-\left((t_1 - \xi_1)^2 + (t_2 - \xi_2)^2\right)} e^{-2i\pi\left((t_1 - \xi_1)^2 - 6(t_1 - \xi_1)(t_2 - \xi_2) - (t_2 - \xi_2)^2\right)} \\
&\quad \times e^{-\left((t_1 - y_1)^2 + (t_2 - y_2)^2\right)} e^{-2i\pi\left((t_1 - \xi_1)^2 - 6(t_1 - \xi_1)(t_2 - \xi_2) - (t_2 - \xi_2)^2\right)} \frac{1}{\mathcal{F}_{\mathbf{M}}[\widetilde{g_{\mathbf{b}}}](\mathbf{w})} \\
&\quad \times e^{2i\pi\left((\xi_1 - 3\xi_2)(\xi_1 - t_1) + (3\xi_1 + \xi_2)(\xi_2 - t_2)\right)} e^{2i\pi\left((y_1 - 3y_2)(y_1 - t_1) + (3y_1 + y_2)(y_2 - t_2)\right)} \\
&\quad \times e^{i\pi\left(-2w_1{}^2 + w_2{}^2\right)} d\mathbf{t} d\mathbf{w} f(\boldsymbol{\xi}) h(\mathbf{y}) d\boldsymbol{\xi} \, d\mathbf{y}.
\end{aligned}
\quad (44)
$$

*Let*

$$I_u^f(\boldsymbol{\xi}) = f(\boldsymbol{\xi}) e^{i\pi\left(\left(-2u_1{}^2 + u_2{}^2\right) + 2(u_1 \xi_2 + u_1 \xi_2 + u_2 \xi_1 - u_2 \xi_2) + \left(\xi_1{}^2 - 6 \xi_1 \xi_2 - \xi_2{}^2\right)\right)},$$

$$I_u^h(\mathbf{y}) = h(\mathbf{y}) e^{i\pi\left(\left(-2u_1{}^2 + u_2{}^2\right) + 2(u_1 y_2 + u_1 y_2 + u_2 y_1 - u_2 y_2) + \left(y_1{}^2 - 6 y_1 y_2 - y_2{}^2\right)\right)}.$$

*Then, we can obtain*

$$(f \star_{\mathbf{M}} h)(\mathbf{x}) = 4e^{-i\pi\left(\left(-2u_1{}^2+u_2{}^2\right)+2(u_1x_2+u_1x_2+u_2x_1-u_2x_2)+\left(x_1{}^2-6x_1x_2-x_2{}^2\right)\right)}$$

$$\times \int_{\mathbb{R}^2}\int_{\mathbb{R}^2}\int_{\mathbb{R}^2}\int_{\mathbb{R}^2} e^{i\pi\left(\left(-2w_1{}^2+w_2{}^2\right)+2(w_1t_2+w_1t_2+w_2t_1-w_2t_2)+\left(t_1{}^2-6t_1t_2-t_2{}^2\right)\right)}$$

$$\times e^{-i\pi\left(\left(-2w_1{}^2+w_2{}^2\right)+2(w_1x_2+w_1x_2+w_2x_1-w_2x_2)+\left(x_1{}^2-6x_1x_2-x_2{}^2\right)\right)}$$

$$\times e^{-\left((t_1-\xi_1)^2+(t_2-\xi_2)^2\right)}e^{-2i\pi\left((t_1-\xi_1)^2-6(t_1-\xi_1)(t_2-\xi_2)-(t_2-\xi_2)^2\right)}$$

$$\times e^{-\left((t_1-y_1)^2+(t_2-y_2)^2\right)}e^{-2i\pi\left((t_1-y_1)^2-6(t_1-y_1)(t_2-y_2)-(t_2-y_2)^2\right)}\frac{1}{\mathcal{F}_{\mathbf{M}}[\widetilde{g_{\mathbf{b}}}](\mathbf{w})}$$

$$\times e^{2i\pi((\xi_1-3\xi_2)(\xi_1-t_1)+(3\xi_1+\xi_2)(\xi_2-t_2))}e^{2i\pi((y_1-3y_2)(y_1-t_1)+(3y_1+y_2)(y_2-t_2))}$$

$$\times e^{i\pi\left(-2w_1{}^2+w_2{}^2\right)}I_u^f(\boldsymbol{\xi})I_u^h(\mathbf{y})d\mathbf{t}d\mathbf{w}d\boldsymbol{\xi}\,d\mathbf{y},$$

$$= 4e^{-i\pi\left(\left(-2u_1{}^2+u_2{}^2\right)+2(u_1x_2+u_1x_2+u_2x_1-u_2x_2)+\left(x_1{}^2-6x_1x_2-x_2{}^2\right)\right)}$$

$$\times \int_{\mathbb{R}^2}\int_{\mathbb{R}^2}\frac{1}{\mathcal{F}_{\mathbf{M}}[\widetilde{g_{\mathbf{b}}}](\mathbf{w})}e^{i\pi\left(-2w_1{}^2+w_2{}^2\right)}$$

$$\times e^{i\pi\left(\left(-2w_1{}^2+w_2{}^2\right)+2(w_1t_2+w_1t_2+w_2t_1-w_2t_2)+\left(t_1{}^2-6t_1t_2-t_2{}^2\right)\right)}$$

$$\times e^{-i\pi\left(\left(-2w_1{}^2+w_2{}^2\right)+2(w_1x_2+w_1x_2+w_2x_1-w_2x_2)+\left(x_1{}^2-6x_1x_2-x_2{}^2\right)\right)}$$

$$\times \left(\int_{\mathbb{R}^2}I_u^f(\boldsymbol{\xi})e^{-\left((t_1-\xi_1)^2+(t_2-\xi_2)^2\right)}e^{-2i\pi\left((t_1-\xi_1)^2-6(t_1-\xi_1)(t_2-\xi_2)-(t_2-\xi_2)^2\right)}\right.$$

$$\times e^{2i\pi((\xi_1-3\xi_2)(\xi_1-t_1)+(3\xi_1+\xi_2)(\xi_2-t_2))}d\boldsymbol{\xi}\Big)\Big(\int_{\mathbb{R}^2}I_u^h(\mathbf{y})$$

$$\times e^{-\left((t_1-y_1)^2+(t_2-y_2)^2\right)}e^{-2i\pi\left((t_1-y_1)^2-6(t_1-y_1)(t_2-y_2)-(t_2-y_2)^2\right)}$$

$$\times e^{2i\pi((y_1-3y_2)(y_1-t_1)+(3y_1+y_2)(y_2-t_2))}d\mathbf{y}\Big)d\mathbf{t}d\mathbf{w}. \tag{45}$$

*For ease of writing, we let*

$$I(\mathbf{t}) = \left(\int_{\mathbb{R}^2}I_u^f(\boldsymbol{\xi})e^{-\left((t_1-\xi_1)^2+(t_2-\xi_2)^2\right)}e^{-2i\pi\left((t_1-\xi_1)^2-6(t_1-\xi_1)(t_2-\xi_2)-(t_2-\xi_2)^2\right)}\right.$$

$$\times e^{2i\pi((\xi_1-3\xi_2)(\xi_1-t_1)+(3\xi_1+\xi_2)(\xi_2-t_2))}d\boldsymbol{\xi}\Big)\Big(\int_{\mathbb{R}^2}I_u^h(\mathbf{y})e^{-\left((t_1-y_1)^2+(t_2-y_2)^2\right)}$$

$$\times e^{-2i\pi\left((t_1-y_1)^2-6(t_1-y_1)(t_2-y_2)-(t_2-y_2)^2\right)}e^{2i\pi((y_1-3y_2)(y_1-t_1)+(3y_1+y_2)(y_2-t_2))}d\mathbf{y}\Big). \tag{46}$$

*According to (4) and (6), the FMT and the inverse of the FMT can be written as*

$$\mathcal{F}_{\mathbf{M}}[I(\mathbf{t})](\mathbf{w}) = \sqrt{2}\int_{\mathbb{R}^2}I(\mathbf{t})e^{i\pi\left(\left(-2w_1{}^2+w_2{}^2\right)+2(w_1t_2+w_1t_2+w_2t_1-w_2t_2)+\left(t_1{}^2-6t_1t_2-t_2{}^2\right)\right)}d\mathbf{t}, \tag{47}$$

$$f(\mathbf{x}) = \mathcal{F}_{\mathbf{M}^{-1}}[\mathcal{F}_{\mathbf{M}}[f](\mathbf{w})](\mathbf{x})$$

$$= \sqrt{2}\int_{\mathbb{R}^n}\mathcal{F}_{\mathbf{M}}[f](\mathbf{w})e^{-i\pi\left(\left(-2w_1{}^2+w_2{}^2\right)+2(w_1x_2+w_1x_2+w_2x_1-w_2x_2)+\left(x_1{}^2-6x_1x_2-x_2{}^2\right)\right)}d\mathbf{w}. \tag{48}$$

*Thus, we obtain*

$$(f \star_{\mathbf{M}} h)(\mathbf{x}) = 4e^{-i\pi\left(\left(-2u_1{}^2+u_2{}^2\right)+2(u_1x_2+u_1x_2+u_2x_1-u_2x_2)+\left(x_1{}^2-6x_1x_2-x_2{}^2\right)\right)}$$

$$\times \int_{\mathbb{R}^2} \frac{e^{i\pi\left(-2w_1{}^2+w_2{}^2\right)}}{\mathcal{F}_{\mathbf{M}}[\widetilde{g_{\mathbf{b}}}](\mathbf{w})} e^{-i\pi\left(\left(-2w_1{}^2+w_2{}^2\right)+2(w_1x_2+w_1x_2+w_2x_1-w_2x_2)+\left(x_1{}^2-6x_1x_2-x_2{}^2\right)\right)}$$

$$\times \left(\int_{\mathbb{R}^2} I(\mathbf{t})e^{i\pi\left(\left(-2w_1{}^2+w_2{}^2\right)+2(w_1t_2+w_1t_2+w_2t_1-w_2t_2)+\left(t_1{}^2-6t_1t_2-t_2{}^2\right)\right)}d\mathbf{t}\right)d\mathbf{w}$$

$$= 2\sqrt{2}e^{-i\pi\left(\left(-2u_1{}^2+u_2{}^2\right)+2(u_1x_2+u_1x_2+u_2x_1-u_2x_2)+\left(x_1{}^2-6x_1x_2-x_2{}^2\right)\right)}$$

$$\times \int_{\mathbb{R}^2} \frac{e^{i\pi\left(-2w_1{}^2+w_2{}^2\right)}}{\mathcal{F}_{\mathbf{M}}[\widetilde{g_{\mathbf{b}}}](\mathbf{w})} \mathcal{F}_{\mathbf{M}}[I](\mathbf{w})$$

$$\times e^{-i\pi\left(\left(-2w_1{}^2+w_2{}^2\right)+2(w_1x_2+w_1x_2+w_2x_1-w_2x_2)+\left(x_1{}^2-6x_1x_2-x_2{}^2\right)\right)}d\mathbf{w}$$

$$= 2e^{-i\pi\left(\left(-2u_1{}^2+u_2{}^2\right)+2(u_1x_2+u_1x_2+u_2x_1-u_2x_2)+\left(x_1{}^2-6x_1x_2-x_2{}^2\right)\right)}$$

$$\times \mathcal{F}_{\mathbf{M}^{-1}}\left[\frac{e^{i\pi\left(-2w_1{}^2+w_2{}^2\right)}}{\mathcal{F}_{\mathbf{M}}[\widetilde{g_{\mathbf{b}}}](\mathbf{w})} \mathcal{F}_{\mathbf{M}}[I](\mathbf{w})\right](\mathbf{x}). \tag{49}$$

*Therefore,*

$$e^{i\pi\left(\left(-2u_1{}^2+u_2{}^2\right)+2(u_1x_2+u_1x_2+u_2x_1-u_2x_2)+\left(x_1{}^2-6x_1x_2-x_2{}^2\right)\right)}(f \star_{\mathbf{M}} h)(\mathbf{x})$$

$$= 2\mathcal{F}_{\mathbf{M}^{-1}}\left[\frac{e^{i\pi\left(-2w_1{}^2+w_2{}^2\right)}}{\mathcal{F}_{\mathbf{M}}[\widetilde{g_{\mathbf{b}}}](\mathbf{w})} \mathcal{F}_{\mathbf{M}}[I](\mathbf{w})\right](\mathbf{x}). \tag{50}$$

*Performing the FMT on both sides of the above formula,*

$$F_{\mathbf{M}}\left[e^{i\pi\left(\left(-2u_1{}^2+u_2{}^2\right)+2(u_1x_2+u_1x_2+u_2x_1-u_2x_2)+\left(x_1{}^2-6x_1x_2-x_2{}^2\right)\right)}(f \star_{\mathbf{M}} h)(\mathbf{x})\right](\mathbf{w})$$

$$= 2\frac{e^{i\pi\left(-2w_1{}^2+w_2{}^2\right)}}{\mathcal{F}_{\mathbf{M}}[\widetilde{g_{\mathbf{b}}}](\mathbf{w})} \mathcal{F}_{\mathbf{M}}[I](\mathbf{w}), \tag{51}$$

*it follows that*

$$e^{-i\pi\left(-2w_1{}^2+w_2{}^2\right)} F_{\mathbf{M}}\left[e^{i\pi\left(\left(-2u_1{}^2+u_2{}^2\right)+2(u_1x_2+u_1x_2+u_2x_1-u_2x_2)+\left(x_1{}^2-6x_1x_2-x_2{}^2\right)\right)}(f \star_{\mathbf{M}} h)(\mathbf{x})\right](\mathbf{w})$$

$$\times \mathcal{F}_{\mathbf{M}}[\widetilde{g_{\mathbf{b}}}](\mathbf{w}) = 2\mathcal{F}_{\mathbf{M}}[I](\mathbf{w}). \tag{52}$$

*According to the convolution theorem of the FMT in (17), the above formula can be rewritten as*

$$\mathcal{F}_{\mathbf{M}}\left[\left[e^{i\pi\left(\left(-2u_1{}^2+u_2{}^2\right)+2(u_1x_2+u_1x_2+u_2x_1-u_2x_2)+\left(x_1{}^2-6x_1x_2-x_2{}^2\right)\right)}(f \star_{\mathbf{M}} h)(\mathbf{x})\right] \circledast_{\mathbf{M}} \widetilde{g_{\mathbf{b}}}\right](\mathbf{w})$$

$$= 2\mathcal{F}_{\mathbf{M}}[I](\mathbf{w}); \tag{53}$$

*thus,*

$$\left[e^{i\pi\left(\left(-2u_1{}^2+u_2{}^2\right)+2(u_1x_2+u_1x_2+u_2x_1-u_2x_2)+\left(x_1{}^2-6x_1x_2-x_2{}^2\right)\right)}(f \star_{\mathbf{M}} h)(\mathbf{x})\right] \circledast_{\mathbf{M}} \widetilde{g_{\mathbf{b}}} = 2I. \tag{54}$$

*The left side of the above equation can be written as*

$$\left(\left[e^{i\pi\left(\left(-2u_1{}^2+u_2{}^2\right)+2(u_1x_2+u_1x_2+u_2x_1-u_2x_2)+\left(x_1{}^2-6x_1x_2-x_2{}^2\right)\right)}(f \star_{\mathbf{M}} h)(\mathbf{x})\right] \circledast_{\mathbf{M}} \widetilde{g_{\mathbf{b}}}\right)(\mathbf{b})$$

$$= \sqrt{2}\int_{\mathbb{R}^2}(f \star_{\mathbf{M}} h)(\mathbf{x})e^{-\left(x_1{}^2+x_2{}^2\right)}e^{i\pi\left(\left(-2u_1{}^2+u_2{}^2\right)+2(u_1x_2+u_1x_2+u_2x_1-u_2x_2)+\left(x_1{}^2-6x_1x_2-x_2{}^2\right)\right)}d\mathbf{x}, \tag{55}$$

*and the right side can be written as*

$$2I(\mathbf{b}) = (\mathcal{G}_g^{\mathbf{M}} f)(\mathbf{b}, \mathbf{u})(\mathcal{G}_g^{\mathbf{M}} h)(\mathbf{b}, \mathbf{u}). \tag{56}$$

*According to the definition of the WFMT (7), we know*

$$(\mathcal{G}_g^{\mathbf{M}}(f \star_{\mathbf{M}} h))(\mathbf{b}, \mathbf{u}) = \sqrt{2} \int_{\mathbb{R}^2} (f \star_{\mathbf{M}} h)(\mathbf{x}) e^{-(x_1{}^2 + x_2{}^2)}$$
$$\times e^{i\pi\left(\left(-2u_1{}^2 + u_2{}^2\right) + 2(u_1 x_2 + u_1 x_2 + u_2 x_1 - u_2 x_2) + \left(x_1{}^2 - 6x_1 x_2 - x_2{}^2\right)\right)} d\mathbf{x}, \tag{57}$$

*which is equal to (55). Thus, we have*

$$(\mathcal{G}_g^{\mathbf{M}}(f \star_{\mathbf{M}} h))(\mathbf{b}, \mathbf{u}) = (\mathcal{G}_g^{\mathbf{M}} f)(\mathbf{b}, \mathbf{u})(\mathcal{G}_g^{\mathbf{M}} h)(\mathbf{b}, \mathbf{u}). \tag{58}$$

*Therefore, we obtain the convolution theorem of the above concrete example, which is consistent with the convolution theorem proposed in this paper.*

## 4. Existence Theorems

In this section, we introduce some existence theorems of the proposed convolution theorem.

**Lemma 1.** *Let $g \in L^2(\mathbb{R}^n)$ be a window function, $f \in L^2(\mathbb{R}^n)$ and $h \in L^2(\mathbb{R}^n)$. Then, the Parseval theorem of the windowed free metaplectic transform is obtained by*

$$\int_{\mathbb{R}^n} \int_{\mathbb{R}^n} (\mathcal{G}_g^{\mathbf{M}} f)(\mathbf{b}, \mathbf{u}) \overline{(\mathcal{G}_g^{\mathbf{M}} h)(\mathbf{b}, \mathbf{u})} d\mathbf{u} d\mathbf{b} = \|g\|_2^2 \langle f, h \rangle_2. \tag{59}$$

**Theorem 3.** *Let $g \in L^2(\mathbb{R}^n)$ be a window function, $f \in L^2(\mathbb{R}^n)$ and $h \in L^2(\mathbb{R}^n)$. Thus, we can obtain*

$$\|f \star_{\mathbf{M}} g\|_2 \leq \|f\|_2 \|g\|_2 \|h\|_2. \tag{60}$$

**Proof of Theorem 3.** For $f \in L^2(\mathbb{R}^n)$ and $h \in L^2(\mathbb{R}^n)$, $\frac{1}{r} + \frac{1}{s} = 1$, $1 \leq r, s < \infty$, we have

$$\|f \circledast_{\mathbf{M}} h\|_\infty \leq \|f\|_r \|h\|_s. \tag{61}$$

Thus,

$$\left| (\mathcal{G}_g^{\mathbf{M}} f)(\mathbf{b}, \mathbf{u}) \right| = \left| \left( \widetilde{f_{\mathbf{u}}} \circledast_{\mathbf{M}} \widetilde{g_{\mathbf{b}}} \right) \right| \leq \|\widetilde{f_{\mathbf{u}}}\|_2 \|\widetilde{g_{\mathbf{b}}}\|_2 = \|f\|_2 \|g\|_2. \tag{62}$$

From Lemma (1), we have

$$\left\| \left( \mathcal{G}_g^{\mathbf{M}} f \right)(\mathbf{b}, \mathbf{u}) \right\|_2 = \|f\|_2 \|g\|_2. \tag{63}$$

So,

$$\left\| \mathcal{G}_g^{\mathbf{M}}(f \star_{\mathbf{M}} h)(\mathbf{b}, \mathbf{u}) \right\|_2 = \left\| \left( \mathcal{G}_g^{\mathbf{M}} f \right)(\mathbf{b}, \mathbf{u}) \left( \mathcal{G}_g^{\mathbf{M}} h \right)(\mathbf{b}, \mathbf{u}) \right\|_2$$
$$= \left( \int_{\mathbb{R}^n} \int_{\mathbb{R}^n} \left| \left( \mathcal{G}_g^{\mathbf{M}} f \right)(\mathbf{b}, \mathbf{u}) \left( \mathcal{G}_g^{\mathbf{M}} h \right)(\mathbf{b}, \mathbf{u}) \right|^2 \right)^{\frac{1}{2}}$$
$$\leq \|f\|_2 \|g\|_2 \left( \int_{\mathbb{R}^n} \left| \left( \mathcal{G}_g^{\mathbf{M}} h \right)(\mathbf{b}, \mathbf{u}) \right|^2 \right)^{\frac{1}{2}}$$
$$= \|f\|_2 \|g\|_2 \left\| \mathcal{G}_g^{\mathbf{M}} h \right\|_2$$
$$= \|f\|_2 \|g\|_2^2 \|h\|_2. \tag{64}$$

Since $\left\| \mathcal{G}_g^{\mathbf{M}}(f \star_{\mathbf{M}} h)(\mathbf{b}, \mathbf{u}) \right\|_2 = \|f \star_M h\|_2 \|g\|_2$, we obtain

$$\|f \star_{\mathbf{M}} h\|_2 \|g\|_2 \leq \|f\|_2 \|g\|_2^2 \|h\|_2. \tag{65}$$

Hence, we obtain

$$\|f \star_M h\|_2 \leq \|f\|_2 \|g\|_2 \|h\|_2.$$

Thus, the proof is finished.    □

**Theorem 4.** *For the free real symplectic matrix* $\mathbf{M}$, *let* $f \in L^r(\mathbb{R}^n)$, $h \in L^s(\mathbb{R}^n)$, $1 < r, s < \infty$, *and* $g \in L^{r'}(\mathbb{R}^n) \cap L^{s'}(\mathbb{R}^n)$ *be a window function,* $\frac{1}{r} + \frac{1}{r'} = 1$, $\frac{1}{s} + \frac{1}{s'} = 1$; *with respect to the free real symplectic matrix* $\mathbf{M} = (\mathbf{A}, \mathbf{B}; \mathbf{C}, \mathbf{D})$, *we have*

$$\left| \mathcal{G}_g^{\mathbf{M}}(f \star_{\mathbf{M}} h)(\mathbf{b}, \mathbf{u}) \right| \leq \|f\|_r \|g\|_{r'} \|h\|_s \|g\|_{s'}. \tag{66}$$

**Proof of Theorem 4.**

$$\begin{aligned}
\left| \mathcal{G}_g^{\mathbf{M}}(f \star_{\mathbf{M}} h)(\mathbf{b}, \mathbf{u}) \right| &= \left| \left( \mathcal{G}_g^{\mathbf{M}} f \right)(\mathbf{b}, \mathbf{u}) \left( \mathcal{G}_g^{\mathbf{M}} h \right)(\mathbf{b}, \mathbf{u}) \right| \\
&= \left| \left( \widetilde{f_{\mathbf{u}}} \circledast_{\mathbf{M}} \widetilde{g_{\mathbf{b}}} \right)(\mathbf{b}) \left( \widetilde{h_{\mathbf{u}}} \circledast_{\mathbf{M}} \widetilde{g_{\mathbf{b}}} \right)(\mathbf{b}) \right| \\
&\leq \|f\|_r \|g\|_{r'} \|h\|_s \|g\|_{s'}.
\end{aligned}$$

Thus, the proof is finished.    □

**Theorem 5.** *Let* $g \in L^2(\mathbb{R}^n)$ *be a window function,* $f, h \in L^2(\mathbb{R}^n)$. *Then,*

$$\left| \int_{\mathbb{R}^n} \int_{\mathbb{R}^n} \left| \mathcal{G}_g^{\mathbf{M}}(f \star_{\mathbf{M}} h)(\mathbf{b}, \mathbf{u}) \right| d\mathbf{u} d\mathbf{b} \right| \leq \|f\|_2 \|h\|_2 \|g\|_2^2. \tag{67}$$

**Proof of Theorem 5.** According to Lemma (1) and the Hölder inequality, we have

$$\begin{aligned}
\left| \int_{\mathbb{R}^n} \int_{\mathbb{R}^n} \left| \mathcal{G}_g^{\mathbf{M}}(f \star_{\mathbf{M}} h)(\mathbf{b}, \mathbf{u}) \right| d\mathbf{u} d\mathbf{b} \right| &= \left| \int_{\mathbb{R}^n} \int_{\mathbb{R}^n} \left| \left( \mathcal{G}_g^{\mathbf{M}} f \right)(\mathbf{b}, \mathbf{u}) \left( \mathcal{G}_g^{\mathbf{M}} h \right)(\mathbf{b}, \mathbf{u}) \right| d\mathbf{u} d\mathbf{b} \right| \\
&\leq \|g\|_2^2 \left| \int_{\mathbb{R}^n} f(\mathbf{t}) h(\mathbf{t}) dt \right| \\
&\leq \|f\|_2 \|h\|_2 \|g\|_2^2.
\end{aligned}$$

Thus, the proof is finished.    □

**Theorem 6.** *Let* $g \in L^q(\mathbb{R}^n)$ *be a window function,* $f, h \in L^p(\mathbb{R}^n)$, $1 \leq p, q < \infty$, $\frac{1}{p} + \frac{1}{q} - 1 \geq 0$, $(f \star_{\mathbf{M}} h) \in L^r(\mathbb{R}^n)$, *and* $\frac{1}{r} = \frac{1}{p} + \frac{1}{q} - 1$. *Then,*

$$\left\| \mathcal{G}_g^{\mathbf{M}}(f \star_{\mathbf{M}} h)(\mathbf{b}, \mathbf{u}) \right\|_r \leq \frac{1}{\sqrt{det\mathbf{B}}} \|f\|_p \|h\|_p \|g\|_q^2. \tag{68}$$

**Proof of Theorem 6.** By (16) and Young's Inequality, we have

$$\begin{aligned}
\|f \circledast_{\mathbf{M}} g\|_r &= \left( \int_{\mathbb{R}^n} \left| \frac{1}{\sqrt{det\mathbf{B}}} (f_1 * g_1)(\mathbf{b}) \right|^r d\mathbf{b} \right)^{\frac{1}{r}} \\
&= \frac{1}{\sqrt{det\mathbf{B}}} \|f_1 * g_1\|_r \\
&\leq \frac{1}{\sqrt{det\mathbf{B}}} \|f_1\|_p \|g_1\|_q, \tag{69}
\end{aligned}$$

where $f_1(\mathbf{x}) = f(\mathbf{x}) e^{i\pi \left( \mathbf{x}^T \mathbf{B}^{-1} \mathbf{A}(\mathbf{x} - \mathbf{b}) \right)}$, $g_1(\mathbf{x}) = g(\mathbf{x}) e^{i\pi \left( \mathbf{x}^T \mathbf{B}^{-1} \mathbf{A}(\mathbf{x} - \mathbf{b}) \right)}$, and $*$ denotes the common convolution operator. It is apparent that $\|f_1\|_p = \|f\|_p$, $\|g_1\|_q = \|g\|_q$; thus,

$$\|f \circledast_{\mathbf{M}} g\|_r \leq \frac{1}{\sqrt{det\mathbf{B}}} \|f\|_p \|g\|_q. \tag{70}$$

Hence,

$$
\begin{aligned}
\left\| \mathcal{G}_g^{\mathbf{M}} f \right\|_r &= \left\| \widetilde{f_{\mathbf{u}}} \circledast_{\mathbf{M}} \widetilde{g_{\mathbf{b}}} \right\|_r \\
&\leq \frac{1}{\sqrt{det\mathbf{B}}} \|\widetilde{f_{\mathbf{u}}}\|_p \|\widetilde{g_{\mathbf{b}}}\|_q = \frac{1}{\sqrt{det\mathbf{B}}} \|f\|_p \|g\|_q.
\end{aligned} \tag{71}
$$

Since $\left| \left( \mathcal{G}_g^{\mathbf{M}} h \right)(\mathbf{b}, \mathbf{u}) \right| = \left| \left( \widetilde{h_{\mathbf{u}}} \circledast_{\mathbf{M}} \widetilde{g_{\mathbf{b}}} \right)(\mathbf{u}) \right| \leq \|h\|_p \|g\|_q$, thus

$$
\begin{aligned}
\left\| \mathcal{G}_g^{\mathbf{M}}(f \star_{\mathbf{M}} h) \right\|_r &= \left\| \left( \mathcal{G}_g^{\mathbf{M}} f \right) \left( \mathcal{G}_g^{\mathbf{M}} h \right) \right\|_r \\
&= \left( \int_{\mathbb{R}^n} \int_{\mathbb{R}^n} \left| \left( \mathcal{G}_g^{\mathbf{M}} f \right)(\mathbf{b}, \mathbf{u}) \left( \mathcal{G}_g^{\mathbf{M}} h \right)(\mathbf{b}, \mathbf{u}) \right|^r d\mathbf{u} d\mathbf{b} \right)^{\frac{1}{r}} \\
&\leq \|h\|_p \|g\|_q \left( \int_{\mathbb{R}^n} \int_{\mathbb{R}^n} \left| \left( \mathcal{G}_g^{\mathbf{M}} f \right)(\mathbf{b}, \mathbf{u}) \right|^r d\mathbf{u} d\mathbf{b} \right)^{\frac{1}{r}} \\
&= \|h\|_p \|g\|_q \left\| \mathcal{G}_g^{\mathbf{M}} f \right\|_r \\
&\leq \frac{1}{\sqrt{det\mathbf{B}}} \|f\|_p \|h\|_p \|g\|_q^2.
\end{aligned}
$$

Thus, the proof is finished. □

## 5. Conclusions

In this paper, according to the definition of the windowed free metaplectic transformation, we introduced a convolution theorem in the WFMT, which has an elegant and simple result as the traditional FT in the frequency domain.This convolution theorem is a perfect extension of the convolution theorem in the WFRFT domain. Finally, according to the new convolution of the WFMT, we introduced several kinds of existence theorems for the convolution theorem of the WFRFT.

**Author Contributions:** Conceptualization, M.C. and Z.Z.; methodology, M.C.; validation, M.C. and Z.Z.; formal analysis, M.C. and Z.Z.; writing—original draft preparation, M.C.; writing—review and editing, M.C. and Z.Z. All authors have read and agreed to the published version of the manuscript.

**Funding:** This research was funded by the National Natural Science Foundation of China under Grant No. 61901223, the Natural Science Foundation of Jiangsu Province under Grant No. BK20190769, the Jiangsu Planned Projects for Postdoctoral Research Funds under Grant No. 2021K205B, and the Jiangsu Province High-Level Innovative and Entrepreneurial Talent Introduction Program under Grant No. R2020SCB55.

**Data Availability Statement:** Not applicable.

**Conflicts of Interest:** The authors declare no conflict of interest.

## Abbreviations

The following abbreviations are used in this manuscript:

| | |
|---|---|
| FT | Fourier transform |
| FRFT | Fractional Fourier transform |
| FRT | Fresnel transform |
| LCT | Linear canonical transform |
| FMT | Free metaplectic transformation |
| WFT | Windowed Fourier transform |
| WFRFT | Windowed fractional Fourier transform |
| WLCT | Windowed linear canonical transform |
| WFMT | Windowed free metaplectic transformation |

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
