# Peer review of "The Convolution Theorem Involving Windowed Free Metaplectic Transform"

_fractalfract, doi:10.3390/fractalfract7040321_

Round 1

Reviewer 1 Report

The paper is clearly written. The problem is not particularly challenging, as the conclusion is ultimately expected due to design of the definitions of FMT and WFMT convolution and the application of basic, standard arguments. The new results could be judged of some interest to a limited community of people working on WMFT. Nevertheless, it should be stressed that any experienced researcher would have no serious problem deriving these results on his own using pretty standard arguments.

Actually, the authors should make an effort to elucidate why the problem considered here is relevant and the reasons behind the introduction of such convolution products. At the moment, it looks like they are designed for the sole purpose of making the convolution product formula work as expected. In any case, the effort of generalization is eventually not repaid by the introduction of new interesting tools or ideas.  

Even if the paper seems accurate from the technical point of view, the recommendation ultimately relates to the fit of the manuscript to the level and scope of the journal. While this evaluation is left to the handling editor, I have no strong reasons to recommend rejection in light of the average quality of the papers appearing in Fractal Fract.

In case of acceptance, authors are encouraged to perform a careful revision of the text due to slight misprints (e.g., "young" in p. 8) and inaccuracies (e.g., the starting line of the proof of Theorem 3 must be fixed).

Also, it could be helpful to the reader to emphasize that (free) metaplectic transforms are nowadays a central tool in time-frequency analysis, with several applications to PDE and mathematical physics. For further details in this connection, we could mention some recent monographs and the references therein:

- de Gosson, Maurice A. Symplectic methods in harmonic analysis and in mathematical physics. Pseudo-Differential Operators. Theory and Applications, 7. Birkhäuser/Springer Basel AG, Basel, 2011.
- Cordero, Elena; Rodino, Luigi. Time-frequency analysis of operators. De Gruyter Studies in Mathematics, 75. De Gruyter, Berlin, 2020.
- Nicola, Fabio; Trapasso, S. Ivan. Wave packet analysis of Feynman path integrals. Lecture Notes in Mathematics, 2305. Springer, Cham, 2022.

Reviewer 2 Report

In this article, the authors have introduced a novel type of convolution for the free metapletic windowed transform and derived the convolution theorem for it. The results have been organized in a nice fashion and are novel in nature. All the results are mathematically correct except with some typos in the proofs. Keeping all this into account, I would like recommend this study for its possible publication in Fractal and Fractional after addressing the following issues:

1. Although the obtained results are mathematically sound, no example has been put forward to validate the results. Authors are advised to present illustrative examples especially for the proposed convolution and the corresponding convolution theorem. 

2. The convolution theorem associated with the free metaplectic LCT has been studied with the other transforms such as [25,29,30]. Authors are requested to explain and discuss briefly how their proposed convolution is different and novel in nature  in comparison to the aforementioned articles.
